# Shiga-Toxin-Producing Strains of *Escherichia coli* O104:H4 and a Strain of O157:H7, Which Can Cause Human Hemolytic Uremic Syndrome, Differ in Biofilm Formation in the Presence of CO_2_ and in Their Ability to Grow in a Novel Cell Culture Medium

**DOI:** 10.3390/microorganisms11071744

**Published:** 2023-07-03

**Authors:** Kei Amemiya, David A. Rozak, Jennifer L. Dankmeyer, William R. Dorman, Charles Marchand, David P. Fetterer, Patricia L. Worsham, Brett K. Purcell

**Affiliations:** 1Bacteriology Division, United States Army Medical Research Institute of Infectious Diseases, Fort Detrick, MD 21702, USA; jennifer.l.dankmeyer.civ@health.mil (J.L.D.); charliemarchanddvm@gmail.com (C.M.); dfed1ster@gmail.com (D.P.F.); worsham@fred.net (P.L.W.); 2Diagnostic Systems Division, United States Army Medical Research Institute of Infectious Diseases, Fort Detrick, MD 21702, USA; david.a.rozak2.civ@health.mil (D.A.R.); william.r.dorman4.civ@health.mil (W.R.D.); 3Department of Medicine, University of Florida, Orlando, FL 32816, USA

**Keywords:** *E. coli* O104:H4, *E. coli* O157:H7, biofilm, novel growth medium

## Abstract

One pathogen that commonly causes gastrointestinal illnesses from the consumption of contaminated food is *Escherichia coli* O157:H7. In 2011 in Germany, however, there was a prominent outbreak of bloody diarrhea with a high incidence of hemolytic uremic syndrome (HUS) caused by an atypical, more virulent *E. coli* O104:H4 strain. To facilitate the identification of this lesser-known, atypical *E. coli* O104:H4 strain, we wanted to identify phenotypic differences between it and a strain of O157:H7 in different media and culture conditions. We found that *E. coli* O104:H4 strains produced considerably more biofilm than the strain of O157:H7 at 37 °C (*p* = 0.0470–0.0182) Biofilm production was significantly enhanced by the presence of 5% CO_2_ (*p* = 0.0348–0.0320). In our study on the innate immune response to the *E. coli* strains, we used HEK293 cells that express Toll-like receptors (TLRs) 2 or 4. We found that *E. coli* O104:H4 strains had the ability to grow in a novel HEK293 cell culture medium, while the *E. coli* O157:H7 strain could not. Thus, we uncovered previously unknown phenotypic properties of *E. coli* O104:H4 to further differentiate this pathogen from *E. coli* O157:H7.

## 1. Introduction

One pathogen commonly associated with small outbreaks of gastrointestinal illnesses from the consumption of contaminated food is *Escherichia coli* O157:H7, which is a Shiga-toxin-producing *E. coli* (STEC) [1,2]. Clinical presentations of *E. coli* O157:H7 infections can include bloody diarrhea, hemorrhagic colitis, and hemolytic-uremic syndrome (HUS) [3,4]. HUS is a sequela of three conditions: hemolytic anemia, thrombocytopenia, and acute renal failure [4,5]. Because antibiotics can exacerbate the toxicity of infections leading to HUS, currently, there are only supportive treatments for patients with STEC-caused bloody diarrhea [6,7,8]. There are two major types of Shiga toxins (Stxs) with subtypes of each produced by *E. coli* O157:H7: Stx1 and Stx2 [9,10]. Based on epidemiological studies of outbreaks and animal model studies, Stx2 appears to be the primary subtype responsible for causing bloody diarrhea and HUS [11,12,13].

Less frequent and smaller outbreaks of food-borne bacterial contamination have been attributed to non-O157:H7 serotypes. However, in the spring of 2011 in Germany, an unusually large number of cases (~3800) of bloody diarrhea and HUS were caused by a less well-known *E. coli* of the serotype O104:H4, with a few cases detected in France and Denmark [14,15,16,17]. The outbreak was attributed to the consumption of contaminated raw sprouts [18]. It was also notable because more adults (~80% of cases, of which two-thirds were women) were affected than children, in contrast to what is usually observed with *E. coli* O157:H7 infections, and there was a higher incidence of HUS (20%) [15]. The *E. coli* O104:H4 strain appeared to exhibit characteristics of two *E. coli* pathotypes: enteroaggregative (EAEC) and enterohemorrhagic (EHEC) [14,15,17,19]. The hallmark of EAEC pathotypes includes the ability to form “stacked brick” aggregates attached to HEp-2 cells which are encoded on a 74 kb virulence plasmid (pAA). The pAA plasmid also expresses a bundle-forming, aggregative adherence fimbriae I (AAF/I), which is in part responsible for this phenotype [20,21,22]. The expression of pAA-encoded genes also enables the pathogen to form biofilms and hemagglutinate human erythrocytes [20,23]. The EHEC pathotypes have the ability to produce attaching and effacing (A/E) lesions in human and animal models [24,25] and can cause hemorrhagic colitis (HC) that can lead to HUS. They also contain a temperate phage that can express heat-stable Stxs [11,13].

After adhering to intestinal epithelial cells *E. coli* O157:H7 and non-O157:H7, strains can initiate biofilm formation. This has been reported to occur in four stages: (1) initial contact, (2) attachment, (3) maturation, and (4) dispersion [26]. In the initial contact with a surface stage, the motility of the microorganism appears to be critical for successful biofilm formation [27,28]. In addition, it was shown that type I pili (or fimbriae) were also required for initial attachment to a surface by *E. coli* [27]. After the initial stage of biofilm formation, attachment of the bacteria to the surface and themselves is mediated by multiple factors, which may include type I pili [27,29], aggregative adherence fimbriae (AAF/I) [30,31,32,33], amyloid curli fibers [34,35,36], long polar fimbriae (lpf) [37,38], and F9 fimbriae [39,40]. Other components that may enhance attachment to the early biofilm are bacterial capsules and LPS [26]. In the maturation stage of biofilm formation, the previous macromolecules are still involved in the process, but there is an increase in bacterial multiplication and bacterial cell-to-cell communication (quorum sensing) [41]. Further secretion of autotransporter adhesins and exopolysaccharides, such as colonic acid, and the synthesis of poly-N- acetyl-glucosamine and possibly cellulose appear to form part of the matrix of the maturating biofilm [26,41]. In the last stage of biofilm formation, which has been less studied, the dispersion of individual bacterial cells or clusters of cells from the mature biofilm may occur by complex mechanisms [42]. These mechanisms can be divided into two general categories: active and passive [42]. In the active category, the mechanism of dispersal is initiated by the bacteria. In the passive category, the mechanism may be mediated by environmental sources, such as fluid movement [42].

Currently, limited information is available about the early interaction of EAEC/EHEC pathotypes with a host’s innate immune response. Part of this germ-line-encoded immune response by a host consists of a class of pathogen recognition receptors (PRR) that are dimerized transmembrane Toll-like receptors (TLRs). TLRs respond to different classes of antigens, which have been described as pathogen-associated molecular patterns (PAMPs) [43]. Humans have 10 TLRs, while mice have 12 [43]. TLRs can be grouped according to the type of PAMPs that they are likely to encounter. Some are assembled on the surface of the host cell and encounter the pathogen early after infection. These consist of TLR1, TLR2, TLR4, TLR5, TLR6, and TLR11; the latter is a pseudogene in humans [44]. Other TLRs, such as TLR3, TLR7, TLR8, and TLR9, are assembled within endosomal compartments in the host cell and activated by nuclei acid components of viral particles or bacterial pathogens. Our interest is in the early innate immune response to gram-negative bacterial pathogens; hence, we have focused our studies on the activation of surfaced-assembled TLRs [45]. We wanted to compare the early interaction of the EHEC/EAEC *E. coli* O104:H and the EHEC *E. coli* O157:H Gram-negative pathogens with the surface-assembled TLR2 and TLR4. For the activation of TLR2 to occur, however, it must combine with TLR1 to form a heterodimer that can recognize triacylated lipopeptides; TLR2 can also dimerize with TLR6 to recognize diacylated lipopeptides [46,47,48,49]. TLR4, on the other hand, is a homodimer that responds to lipopolysaccharides (LPS). TLR4 also requires accessory proteins, such as the LPS binding protein (LBP), myeloid differentiation factor 2 (MD2), and CD14, for optimal activity [46,50,51]. TLR5 is a homodimer and can be activated by flagella or its flagellin subunit [45,52,53,54].

Since less is known about the atypical *E. coli* O104:H4 pathotype than the more common *E. coli* O157:H7, we wanted to identify additional phenotypic differences between these two HUS-causing STECs. We directly compared the two pathogens under different growth or culture conditions to facilitate the identification of the atypical *E. coli* O104:H4 versus the O157:H7 pathogen. We report here that CO_2_ significantly enhanced biofilm formation by the *E. coli* O104:H4 strains at 37 °C but not by the *E. coli* O157:H7 strain. In our examination of the interaction of the *E. coli* strains with a host’s innate immune response using HEK293 cells that expressed TLR2 or 4, we found that the *E. coli* O104:H4 strains but not the O157:H7 strain had the ability to grow in a novel TLR cell culture medium.

## 2. Materials and Methods

### 2.1. Bacterial Strains, Reagents, Growth Conditions, and Antibiotic Susceptibility

The *E. coli* strains used in the present study are listed in Table 1 and were obtained from the Critical Reagent Program (CRP) at the U.S. Army Medical Research Institute of Infectious Diseases (USAMRIID, Fort Detrick, MD, USA). For ease of labeling in the present report, the German (*E. coli* O104:H4, 2011C 34923) and two Georgian (*E. coli* O104:H4, 2009EL 2050 and *E. coli* O104:H4, 2009 EL 2071) strains listed in Table 1 will be referred to as Ec 01, Ec 02, or Ec 03, respectively. Stock cultures of strains were stored at −80 °C in 50% glycerol in Mueller-Hinton (MH) II broth, cation-adjusted (BD Difco, Thermo Fisher Scientific, Waltham, MA, USA).

For growth studies, stock cultures were streaked out onto sheep-blood agar (SBA) (Remel, Thermo Fisher Scientific) or MH II agar (1.5%) (BD Difco, Thermo Fisher Scientific) plates, and the plates were incubated overnight at 37 °C. A single colony was used to inoculate 2–5 mL of cation-adjusted MH II broth, and the culture was incubated at 37 °C overnight with shaking at 180 rpm. The following day, the overnight culture was diluted 1/50–1/100 in MH II broth or other desired media for further studies. Alternatively, fresh overnight colonies on a solid medium were used to make a suspension of cells for an inoculum in 96-well plates or 5 mL tubes. For the quantitation of *E. coli* bacterial cells, the conversion factor of 1.0 OD_600_ nm, equivalent to 6.83 × 10^8^ colony-forming units (CFUs) was used (Henry Heine, USAMRIID). To determine CFUs in desired cultures, 10-fold dilutions of the culture were made in 1X PBS, and 100 µL of each dilution was plated onto triplicate SBA plates. The SBA plates were incubated at 37 °C overnight and the number of colonies was counted to determine the original CFU/mL.

In our initial assessment of the *E. coli* strains used in the present study, we used Biolog GEN III microplates (Biolog, Hayward, CA, USA) to obtain a phenotypic display of common or differing characteristics. After bacterial strains were grown overnight on SBA plates at 37 °C, bacterial cells and GEN III microplates (carbon utilization) were prepared as per the manufacturer’s directions. We also evaluated the antibiotic susceptibilities of the *E. coli* strains used in the present study. This was determined as previously described by the broth dilution method following the Clinical and Laboratory Standards Institute method [55]. The results from the Biolog GENIII plates and antibiotic susceptibilities studies of the *E. coli* strains are shown in Appendix A, respectively, and in part in Table 1. Generally, the *E. coli* O104:H4 strains exhibited the same phenotypes or antibiotic susceptibilities. The results for the O157:H7 strain were either the same or different from the O104:H4 strains depending on the carbohydrate/chemical or antibiotic being examined. To further characterize the *E. coli* strains, we examined the phenotype of the *E. coli* strains on MacConkey agar with sorbitol and a tryptone bile x-glucuronide medium (Remel, ThemoFisher Scientific). To examine the motility of the *E. coli* strains, we inoculated MH II agar (0.4%) plates with an overnight culture of the strains. The plates were examined for bacterial motility after incubation overnight at 37 °C. Difco Heart Infusion agar (ThermoFisher Scientific, Waltham, MA, USA) for Congo Red (Sigma-Aldrich, St. Louis, MO, USA) (100 µg/mL) plates were used for dye utilization/uptake studies (see Appendix A) [56]. The presence of Stx2 (List Labs, Campbell, CA, USA) was determined by Western slot-blot analysis of filtered culture supernates with an anti-Stx2 monoclonal antibody [57] that was obtained from Dr. Alison D. O’Brian (Department of Microbiology and Immunology, Uniformed Services University of the Health Sciences, Bethesda, MD, USA). The results of these studies and other characteristics [58,59,60] are presented in Table 1.

### 2.2. Biofilm Formation

Biofilm formation was examined in 96-well round-bottom plates (Corning Falcon, ThermoFisher) and 5 mL polypropylene tubes (12 × 75 mm, Corning Falcon, Thermo Fisher). Bacterial strains were grown on SBA plates overnight at 37 °C, and cells were suspended in MH II broth and OD_600_ adjusted to 1.0, as stated previously. For 96-well ELISA plates, 10–20 µL of adjusted cell suspension was used to inoculate 180–190 µL of MH II broth or 200–500 µL MH II broth in 5 mL cultures tubes with snap-cap lids (to safely wash and process the pellets). Bacterial cultures were incubated overnight at 37 °C with or without 5% CO_2_. When cultures were incubated in 5 mL culture tubes, the lids were loosely closed to allow for air/CO_2_ exchange. Biofilm formation was quantitated in the 5 mL tubes using a modified method [61] with 0.1% crystal violet (in water and filtered through a 0.45 µm filter) (Sigma-Aldrich). All washing and processing of biofilm pellets by *E. coli* strains were performed under a biosafety hood under Biosafety Level 2 conditions. All centrifugation steps were conducted at 2000 rpm for 5 min in covered buckets, and buckets were loaded or emptied under the biosafety hood. After overnight incubation, 1.0 mL of PBS (ThermoFisher) was added to each sample, and samples were vortexed and centrifuged. A tube with only MH II medium was included as a blank/reagent control. The samples were decanted and washed twice with 1.0 mL of PBS. After the last centrifugation step, 0.1–0.2 mL of crystal violet was added to each sample, including the blank/reagent control, and samples were left at room temperature for 15–30 min. Then, 1.0 mL of PBS was added to each sample, and the samples were vortexed and centrifuged. After centrifugation, 2.0 mL of PBS was added to each sample, and the samples were vortexed. This wash step was repeated twice. After the last wash, the tubes were left to dry under the biosafety hood for at least 3 h. To the dry pellets, 1.0 mL of 95% ethanol was added, and the samples were vortexed vigorously to dissolve the pellets and then left for 15–30 min. In some cases, the samples were left in the solvent overnight with the caps of the tubes snapped closed to ensure complete dissolution of the pellet. Then, 200 µL of each sample was transferred to a 96-well ELISA plate, which included the blank/reagent control sample, and the plates were read at 600 nm. The absorbance of the blank/reagent control sample (absorbance at 600 nm < 0.10) was not subtracted from the readings from the wells that contained the bacterial cells, but it was plotted with the results of the test samples.

### 2.3. Human Embryonic Kidney (HEK) 293 Toll-like Receptor (TLR) Cell Cultures

HEK293 cells expressing TLR2 or 4 used for the evaluation of TLR activation by *E. coli* strains were obtained from Invivogen (San Diego, CA, USA). The TLR-expressing HEK293 cells were also transfected with an embryonic alkaline phosphatase reporter gene. The expression of the reporter gene was dependent on the binding of multiple NF-kB and AP-1 proteins in the enhancer region of the reporter gene. Once expressed, the alkaline phosphatase was secreted (SEAP) into the HEK Blue Detection (HBD) cell culture medium, where its activity could be detected (Invivogen). SEAP cleaved the color indicator 5-bromo4-chloro-3-indolyl phosphate in the HBD cell culture medium. Once the color indicator was cleaved by SEAP, a blue-purple product was generated. The growth and maintenance of the HEK293 TLR2/4 cells and the examination of TLR2/4 activation by live bacterial strains were as previously described [45]. The activation of TLR2/4 by *E. coli* strains in the HBD cell culture medium (Invivogen) was performed in 96-well flat-bottom Immunlon 2HB plates (ThermoFisher, Waltham, MA, USA). The HBD cell culture medium was prepared as per the manufacturer’s directions. Briefly, the contents of a single pouch of HBD powder were poured into a 125 mL sterile flask (Corning, Fisher Scientific), and 50 mL of endotoxin-free water was added to solubilize the powder. The dissolved HBD medium was incubated at 37 °C for 30–60 min with occasional stirring and passed through a 0.22 µm filter. The sterile medium was stored at 4 °C until use for up to 2 weeks. HEK293 TLR2 and TLR4 cells were adjusted to 2.5 × 10^5^ cells/mL and 2.0 × 10^5^ cells/mL, respectively, in an HBD cell culture medium, as per the manufacturer’s directions before use. In the activation studies, 180 µL of adjusted HEK293 TLR2 or 4 cells were inoculated in triplicate wells with 2.5 × 10^4^ bacterial cells per well. For the control wells, 2 × 10^6^ heat-killed *Listeria monocytogenes* (HKLM) (Invivogen) was used for the TLR2 positive control and 0.2 ng *E. coli* lipopolysaccharide (LPS) (Invivogen) was used for the TLR4 positive control. The inoculated HEK293 TLR plates were incubated at 37 °C with 5% CO_2_ overnight for 18–24 h. The results were read at 630 nm with a BioTek Elx808 spectrophotometer (BioTek, Winooski, VT, USA). Similar *E. coli* cultures were performed in an HBD cell culture or DMEM cell culture maintenance medium without HEK293 TLR cells in 96-well plates or 15 mL culture tubes. All readings of cultures were used to evaluate the amount of TLR2 or 4 activation by live bacterial cells in 96-well plates and were used to plot the data, including the media-only wells, which contained only media, unless noted. The latter readings (<0.10 absorbance at 630 nm) were not subtracted from the wells containing *E. coli* cells, but they were plotted with the readings of the other wells.

### 2.4. E. coli Strains in an HBD Cell Culture Medium

To examine the growth of the *E. coli* strains in the HBD cell culture medium, single colonies from SBA plates were used to inoculate 2 mL of MH II broth, and the cultures were incubated overnight at 37 °C with shaking at 180 rpm. The following day, 0.25 mL of the overnight culture was used to inoculate 12.5 mL of the HBD cell culture medium (1/50 dilution) in a 125 mL Corning flask (Fisher Scientific), and the cultures were incubated at 37 °C with shaking at 180 rpm. Growth of the cells was followed by an increase in absorbance at 600 nm for up to 48 h. The HBD cell culture medium was used as a blank for the culture readings. For comparison, *E. coli* strains were also grown in 12 mL of MH II broth at 37 °C with shaking at 180 rpm after inoculation with 0.12 mL of an overnight culture (1/100 dilution), and the growth was followed by an increase in absorbance at 600 nm. MH II broth was used as a blank for the culture readings. In other cases, where noted, single colonies of *E. coli* strains grown on SBA plates were directly used to inoculate 2 mL of the HBD culture medium, and the cell cultures were incubated overnight at 37 °C with 5% CO_2_.

### 2.5. Comparison of Amino Acid Sequence of AAF/I Protein Components

The amino acid sequence of the AggA, AggB, AggC, and AggD proteins for AAF/I was obtained from the National Center for Biotechnology Information (NCBI). For *E. coli* O104:H4 2011C-3493, the amino acid sequence for plasmid pAA proteins was under NCBI reference sequence NC_018666.1. For strain 2009EL-2050, the amino acid sequence for plasmid pAA proteins was under NCBI reference sequence NC_018654.1. For strain 2009EL-2071, the amino acid sequence for plasmid pAA proteins was under NCBI reference sequence NC_018662.1.

### 2.6. Statistical Analysis

Values were log transformed prior to analysis, with results summarized as the geometric mean and geometric standard error of the mean. Comparisons between groups were made by Welch’s *t*-test, applied to the log-transformed values. The comparisons between groups were based on a Welch’s *t*-test of the appropriate single degree of freedom contrast in the two-way ANOVA, with factors corresponding to treatment groups. Analysis was performed in SAS^®^ version 9.4 (SAS Institute Inc., Cary, NC, USA). The results were considered significant when *p* < 0.05.

## 3. Results

### 3.1. Enhanced Biofilm Formation by E. coli O104:H4 in 5% CO_2_

We found clear differences in biofilm formation among *E. coli* strains grown during our studies and identified differences in growth between *E. coli* O104:H4 and O157:H7. We grew the *E. coli* strains in MH II broth in 96-well plates. We noticed a difference in their pellets after overnight incubation at 37 °C (Figure 1A, top panel). *E. coli* O157:H7 and the control *E. coli* ATCC 25922 had well-defined circular pellets, while the *E. coli* O104:H4 strains Ec 01, Ec 02, and Ec 03 had pellets that covered the bottom of the wells. In some instances, closer examination of the pellets of the *E. coli* O104:H4 strains revealed small pocked-marked clearings on the peripheral of the pellets reminiscent of phage plaques (Appendix A). We also found that incubation of the 96-well plates at 37 °C with 5% CO_2_ noticeably altered the appearance of the pellets of the *E. coli* O104:H4 strains. They appeared as wrinkled or folded membranous structures. However, this was not observed in the pellets of the *E. coli* ATCC 25922 or O157:H7 strains (Figure 1A, bottom panel).

We used a crystal violet assay to quantitate the change in pellet formation by the *E. coli* strains with or without CO_2_. *E. coli* strains were grown in an MH II medium (0.2–0.5 mL) in 5 mL culture tubes and cultures were incubated overnight at 37 °C with or without CO_2_. Figure 1B,C present representative results from one of three studies we conducted under these conditions. The pellets of the *E. coli* strains incubated in the 5 mL tubes looked similar to those observed in the 96-well plates with and without CO_2_ (Figure 1B). However, we did not observe plaque-like clearings as we did in the 96-well plates. Figure 1C shows that there was more pellet produced by the *E. coli* O104:H4 strains than the *E. coli* ATCC 25922 or O157:H7 strains under either condition. There was a significant difference in the amount of pellet formed by the *E. coli* O104:H4 strains in the presence of CO_2_ compared to its absence: Ec 01 (*p* = 0.0326), Ec 02 (*p* = 0.0022), and Ec 03 (*p* = 0.0498) (Figure 1C). This difference in the amount of pellet produced in the absence or presence of CO_2_ by *E. coli* O104:H strains was also supported when we compared the amount of pellet formed without CO_2_ between the *E. coli* O157:H7 and O104:H4 strains. The latter strains produced more than 3- to 7-fold more pellets than *E. coli* O157:H7 (Ec 01, *p* = 0.0270; Ec 02, *p* = 0.0470; Ec 03, *p* = 0.0182). When we compared the amount of pellet produced by the *E. coli* O157:H7 and O104:H4 strains grown in the presence of CO_2_, the difference in the amount of pellet formed increased to 10- to 15-fold (Ec 01, *p* = 0.0348; Ec 02, *p* = 0.0320; Ec 02, *p* = 0.0343). Because of previous reports linking *E. coli* O104:H4 with biofilm formation [34,62,63], the pellets produced by the *E. coli* O104:H4 strains will henceforth be referred to as biofilms in this report.

### 3.2. Interaction of the Live E. coli Strains with Innate Immune TLR2 and 4

We used HEK293 cells that expressed TLR2 or 4 to evaluate whether they interacted with the *E. coli* strains. Besides examining differences in the growth between the pathogenic *E. coli* O104:H4 strain and the O157:H7 strain, we also asked if there were possible differences in *E. coli* O104:H4 and O157:H7 interactions with a host’s innate immune system. is activated by acylated lipoproteins, while TLR4 is activated by lipopolysaccharides [46]. The activation of TLRs stimulates the expression of NF-ebb that is required for the enhanced expression of the reporter protein SEAP from stimulated HEK293 cells. This gives a blue/purple color after cleavage of the indicator 5-bromo-4-chloro-3-indolyl phosphate by secretory alkaline phosphatase. We inoculated HEK293 TLR2 or 4 cells in a cell culture medium in 96-well plates (triplicate wells) with the *E. coli* strains and incubated the cultures at 37 °C with 5% CO_2_ overnight. Representative results from one of three TLR2 and 4 activation studies are presented. The results of this study are shown in Figure 2A and are representative results from one of three studies we conducted to examine the activation of TLR2 or 4 by the *E. coli* strains. We observed a notable significant difference in the amount of TLR4 activated compared to the amount of TLR2 activated (*p* = 0.0041–0.0001) by all *E. coli* strains that we evaluated except *E. coli* O157:H7 (Figure 2B). Nevertheless, the *E. coli* O157:H7 strain in this case still activated more TLR4 than TLR2, but the difference in the activation between the two TLRs by *E. coli* O157:H7 was not significant.

When we compared the amount of TLR2 or 4 activated by the *E. coli* O157:H7 strain with the amount activated by the other *E*. *coli* strains, we found that *E. coli* O157:H7 activated significantly less TLR2 (*p* = 0.0114–0.0007), but there was no difference in TLR4 activation.

### 3.3. Growth of the E. coli O104:H4 Strains but Not the O157:H7 Strain in an HBD Cell Culture Medium

When examining the activation of TLR2/4 by the *E. coli* strains, we found a color reaction difference between the *E. coli* ATCC 25922 and O157:H7 strains (blue-purple) and the *E. coli* O104:H4 Ec 01, Ec 02, and Ec 03 strains (yellow-green) (see Figure 2A). To determine if the difference depended on live *E. coli* strains, we inoculated HEK293 TLR2/4 cells with irradiated-inactivated *E. coli* strains. After overnight incubation, all the *E. coli* strains produced a similar blue-purple color reaction, with a higher amount of TLR2 than TLR4 activated. These results suggested that the yellow-green color reaction generated by the *E. coli* O104:H4 strains with HEK293 TLR2/4 cells was dependent on live bacterial cells. We also examined other cell lines, such as RAW 264.7 and Vero cells in an HBD cell culture medium with the live *E. coli* strains. We observed the same color reaction that we observed with *E. coli* strains using HEK293 TLR2/4 cells. To determine if the presence of TLRs on the HEK293 TLR2/4 cells was responsible for the different color reactions, we inoculated HEK293 TLR null cells that did not express TLR2 or 4 in an HBD cell culture medium with live *E. coli* strains. Figure 2C presents representative results from one of three studies we conducted with HEK293 TLR null cells. We observed no activity with the *E. coli* ATCC 25922 or O157:H7 strains or the positive controls, but the wells containing the *E. coli* O104:H4 strains were greenish in color after incubation (Figure 2C). Because the yellow-greenish color reaction by the *E. coli* O104:H4 strains in the HBD cell culture medium did not appear to depend specifically on HEK293 cells or the presence of TLR2 or 4, we inoculated the HBD cell culture medium (without the HEK293 cells) with the *E. coli* strains. This study was performed three times and Figure 2D is representative of one of these studies. We observed no color change with the *E. coli* ATCC 25922 or O157:H7 strains, but we observed a yellowish color reaction with the *E. coli* O104:H4 strains (Figure 2D). Microscopic examination of the wells containing the *E. coli* O104:H4 strains revealed the presence of small rod-shaped, bacterial-like cells. These cells were absent in wells that contained the *E. coli* ATCC 25922 or O157:H7 cells. Thus, we made dilutions of all the wells and plated aliquots onto SBA plates. After incubation overnight at 37 °C, we detected more than 1.0 × 10^8^ CFU/mL from the *E. coli* O104:H4 cultures, but we detected no CFUs from the *E. coli* ATCC 25922 or O157:H7 cultures. This color reaction difference between the *E. coli* strains was not observed with a DMEM cell maintenance medium that is normally used to maintain the HEK293 TLR2/4 cells (Figure 2D). Taken together, these results suggest that the *E. coli* O104:H4 strains but not the *E. coli* ATCC 25922 or O157:H7 strains can grow in an HBD cell culture medium independent of HEK293 TLR2/4 cells.

To further examine the potential of *E. coli* O104:H4 strains to grow in the HBD cell culture medium, we examined cultures of *E. coli* O104:H4 Ec 01, Ec 02, and Ec 03 strains and the *E. coli* ATCC 25922 and O157:H7 strains in 125 mL flasks containing an HBD cell culture medium (12.5 mL). We inoculated the HBD cell culture medium (0.25 mL) with an overnight culture of the *E. coli* strains grown in an MH II medium, and the cultures were incubated at 37 °C with shaking at 180 rpm. We followed growth by tracking the change in absorbance (A) 600 nm, the cell cultures were followed for 48 h. Figure 3A presents representative results from one of three growth studies of the *E. coli* strains in HBD cell culture media in 125 mL flasks. After 48 h, the *E. coli* O104:H4 cultures reached an A_600_ of 1.29–1.38, and we detected 1.6 × 10^8^–4.3 × 10^8^ CFUs/mL present in the *E. coli* O104:H4 cultures. The *E. coli* O104:H4 Ec 03 strain appeared to grow slightly slower than the *E. coli* O104:H4 Ec 01 and Ec 02 strains at the beginning, but it reached close to the same density as the other two *E. coli* O104:H4 strains after 48 h. We observed no change in culture density nor did we recover any CFUs from the *E. coli* ATCC 25922 or O157:H7 cultures after the same time period. Figure 3B shows the color of the *E. coli* cultures in the HBD cell culture medium after 48 h. The *E. coli* O104:H4 Ec 01, Ec 02, and Ec 03 cultures were green, while the *E. coli* ATCC 25922 and O157:H7 cultures remained pink (unchanged). In contrast, when we grew the five *E. coli* strains in MH II broth (12.0 mL) inoculated with 0.12 mL of an overnight culture, we observed that all *E. coli* strains grew at an almost identical rate. They reached a similar A_600_ (~1.80) after 7 h of incubation at 37 °C with shaking at 180 rpm (Figure 3B). The cell densities of the cultures were 5.8 × 10^9^–1.2 × 10^10^ CFU/mL after 7 h of incubation, which was ~10-fold higher than the growth of the *E. coli* O104:H4 strains in the HBD cell culture medium. It should be noted that no supplemental CO_2_ was present during the incubation of these cultures. Thus, the growth of the *E. coli* O104:H4 strains in the HBD cell culture medium was shown by an increase in cell density and a change in the color of the medium.

### 3.4. Growth of the E. coli O104:H4 Strains in an HBD Cell Culture Medium Starting with a Single Colony

We further examined our observation that the *E. coli* O104:H4 strains but not the *E. coli* ATCC 25922 or O157:H7 strains could grow in an HBD cell culture medium. We took single colonies of the five *E. coli* strains grown overnight on SBA plates and inoculated each into 2 mL of an HBD cell culture medium in 15 mL culture tubes. The cultures were then incubated at 37 °C with 5% CO_2_ overnight (without shaking). A single overnight *E. coli* colony on sheep blood agar (~2.5 mm diameter) consists of ~2 × 10^8^ CFUs (Amemiya, unpublished). Figure 4 presents representative results from one of the three independent studies we conducted. After overnight incubation, the *E. coli* O104:H4 Ec 01, Ec 02, and Ec 03 cultures were greenish in color with occasional bubbles on top of the liquid culture. The *E. coli* ATCC 25922 and O157:H7 cultures were light blue-purple in color with no bubbles. The media control stayed pinkish. We checked the pH of the cultures to determine if any metabolic activity occurred. The media control and *E. coli* ATCC 25922 and O157:H7 samples had a pH between 8.5 and 9.0, while the *E. coli* O104:H4 cultures had a pH of ~5.0. The color change in the *E. coli* ATCC 25922 and O157:H7 cultures when compared to the media-only sample may have been due to the presence of alkaline phosphatase, which is found in the periplasmic space of gram-negative bacteria. It may have leaked from the bacterial cells in the original inoculum [63,64]. Thus, we were able to repeat the color reaction change and growth of the *E. coli* O104:H4 cultures that we observed in the larger HBD cell cultures in a different format to differentiate their growth from the *E. coli* O157:H7 strain.

## 4. Discussion

In our observations on the biofilms produced by the EHEC/EAEC pathotype *E. coli* O104:H4, we found that biofilms produced by the O104:H4 strains could be affected by the presence of CO_2_. In our initial study, there was a 3- to 7-fold difference in the substantial amount of biofilm produced by the *E. coli* O104:H4 strains compared to the EHEC pathogen *E. coli* O157:H7 in the absence of CO_2_. We observed a significant enhancement (10- to 15-fold) of the amount of biofilm formed by *E. coli* O104:H4 when grown in the presence of CO_2_. We observed no response of the O157:H7 strain. The effect of CO_2_ on biofilm formation and the response to changes in CO_2_ levels in local environments were found to be critical in some pathogens’ ability to upregulate the expression of virulence factors required to successfully infect a mammalian host [65]. The early stages of biofilm formation begin with the ability of *E. coli* O104:H4 to adhere to and colonize intestinal epithelial cells. This is mediated in part by the biosynthesis of at least two types of fimbriae, which are long, extracellular, polymerized protein structures: AAF from the 74 kb pAA virulence plasmid and LpF from the pathogen’s chromosomal genes [31,33,38]. The presence of AAF has been shown to be required for adherence and biofilm formation by *E. coli* O104:H4 [20,32,62]. It has been suggested that adherence of the pathogen to epithelial cells facilitates the translocation of Stx2a to the intestinal epithelium [66,67]. The expression of genes that encode for AAF is under the control of the AggR regulator, which is also encoded on the pAA plasmid. AggR belongs to the family of AraC-like transcriptional activators and regulates the expression of select genes on the pAA plasmid and EAEC chromosome [31,68]. The upregulation of AggR expression appears to be (p)ppGpp-dependent [69]. There are at least five subtypes of AAF (I–V), with their operons organized as *aggDCBA*, where the *D* gene encodes a chaperone, the *C* gene encodes an usher protein, the *B* gene encodes a minor pilin, and the *A* gene encodes the major pilin subunit of the AAF fimbriae [33,70]. Thus, the assembly of AAF appears to occur through the chaperone–usher pathway that is ubiquitous in Gram-negative bacteria [71]. In a study to evaluate the effect of AAF types (I, III, IV, and V) in an EAEC 55989 agg3^−^ deletion mutant, it was reported that the AAF/I recombinant formed the largest bacterial aggregates and adhered more to Hep-2 cells than the EAEC 55989 agg3^−^ recombinants expressing III, IV, or V fimbriae. These results suggested that the AAF/I subtype may be partly responsible for the extreme pathotype of the outbreak *E. coli* O104:H4 strain [33]. However, the AAF/I-expressing recombinant also exhibited the highest auto-aggregation rate and the lowest biofilm formation of the AAF III, IV, or V recombinants [33]. It was not clear why there was no correlation between an increase in adherence of the different recombinants to Hep2 cells and the amount of biofilm formed with the expression of AAF/I. It had been previously shown that the presence of AAF/I coincides with the ability to form biofilms in *E. coli* O104:H4 [32,62,68]. It may be that biofilm formation is a more temporal, controlled process that requires the biosynthesis of certain molecules in an ordered manner. Biofilm formation may not start with an aggregate of cells formed in a haphazard manner. Additionally, in an EAEC 55989 agg3^−^ bacteria, AAF/I may not mediate biofilm formation, nor in the LB medium used in this study. A recent report on EAEC strains also found a variant AAF (agg3A and agg5A genes) that coincided with the presence of a novel pAA variant [72].

To identify the type of AAF present in the *E. coli* O104:H4 strains used in the present study, we examined the annotated amino acid proteins of the pAA plasmids from the *E. coli* O104:H4 strains 2011C-3493 (Ec01), 2009EL-2050 (Ec02), and 2009E-2071 (Ec03) that had been deposited in GenBank [59]. They are annotated under the NCBI reference sequence nos. NC_018666.1, NC_018654.1, and NC_018662.1, respectively. All the protein subunits AggA, AggB, AggC, and AggD were identified as the aggregative adherence fimbriae I (AAF/I) type. To determine how close the AAF/I subunits were to each other from the different *E. coli* O104:H4 strains, we compared the amino acid sequence of the AggABCD protein subunits from 2011C-3493, 2009E-2050, and 2009E-2071 (see Appendix A). The alignment of the individual subunits showed that there was 100% identity in the amino acid sequences of the AggA, AggC, and AggD proteins. In AggB, there was only one amino acid difference out of 145 amino acids, which was at amino acid 95 (G or W). We also compared the amino acid sequences of the AggR regulator (265 amino acids in length) from the three pAA plasmids. We found them to be identical. In all three pAA plasmids, the *aggR* gene was 10,359 nucleotides from the *aggA* gene. Since AggR positively regulates the expression of the AAF/I operon, we proposed that the direction of transcription is likely to be *→ aggABCD*, which we found in the arrangement for the AAF/I subunit proteins in the 2011C-3493 and 2009EL-2071 strains. In these two cases, the AggR was upstream from the *aggABCD* operon. However, for strain 2009EL-2050, the AAF/I subunits were reported as DCBA; yet, AggR was downstream from the *aggABCD* operon. In this latter case, the direction of transcription may be *DCBAagg* ←. Bielaszewska et al. (2011) had previously reported the presence of *aggA* (AAF/I) in 80 out of 80 *E. coli* O104:H4 strains examined and noted the presence of agg3A in EAEC 55989 [66]. Similarly, Ferdous et al. (2015) identified aggA (AAF/I) in *E. coli* O104:H4 strains, including 2009EL-2050 and 2009EL-2071 and agg3A (AAF/III) in EAEC 55989 [73]. Additionally, Berger et al. (2016) reported the presence of aggA, aggB, aggC, and aggD genes for AAF/I on the pAA plasmid from *E. coli* O104:H4 LB26692 [31]. Thus, the *E. coli* O104:H4 strains used in our study appear to express AAF/I fimbriae and may be in part responsible for the robust biofilm formation by the *E. coli* O104:H4 strains observed in our studies.

Other fimbriae involved in biofilm formation included *lpf,* which was first noted in the outbreak *E. coli* O104:H4 strains by Bielaszewska et al. (2011) [66] and Rasko et al. (2011) [21]. It was subsequently found that the *E. coli* O104:H4 strains had two *lpf* operons (*lpf1* and *lpf2*). They were highly homologous to the *lpf1* and *lpf2* genes in the EAEC 55989 strain but not in the *E. coli* O157:H7 EDL933 strain [38]. The *lpf1* operon consists of genes *A*–*E*, where *lpf1A* is the major fimbriae subunit, *lpf1B* is a chaperone, *lpf1C* is an outer membrane usher protein, *lpf1D* is a minor fimbrial subunit, and *lpf1E* may be a regulator or another fimbrial subunit [38]. The *lpf2* operon consists of genes *A*–*D*, with the possible presence of a duplicate *D* gene, and the protein products are analogous to those found in the *lpf1* operon [38]. Thus, like AAF/I, the assembly of the LpF fimbriae appears to be carried out through the chaperone–usher pathway [71].

It had been previously noted that biofilm formation by *E. coli* O104:H4 was greater than that by *E. coli* O157:H7, both in vitro and in vivo [62]. The in vitro studies, however, were performed without CO_2_ in DMEM with 0.45% glucose. It was reported that *E. coli* O104:H4 produced 7.3-fold greater amounts of biofilm compared with the *E. coli* O157:H7 strain. The authors also examined biofilm formation in a flow cell in DMEM after 16 h and found that biofilm formation only occurred with *E. coli* O104:H4 and not with *E. coli* O157:H7 under these conditions [62]. We did not observe biofilm formation by the *E. coli* O104:H4 strains in an HBD cell culture medium, only in an MH II medium, nor did we observe biofilm formation by the *E. coli* O104:H4 strains in a DMEM maintenance medium, which we used to maintain the HEK293 TLR2/4 cells. However, the maintenance medium also contained other proprietary inhibitory agents. Additionally, our finding that *E. coli* O157:H7 was a weak biofilm producer was previously noted for most isolates of this EHEC strain [28,74,75,76]. Weak biofilm and curli fimbriae production also coincided with the weak binding of Congo Red dye [74,77]. Chen et al. (2013) [28] proposed that there are three factors that influence biofilm formation in *E. coli* strains: (1) prophage insertion into the MerR-like (*mlrA* or previously labeled *yehV*) regulator gene; (2) mutation in RpoS, the stress-response sigma factor required for the expression of stress response genes; and (3) impaired bacterial motility. MlrA and RpoS are both required for biofilm formation and curli fimbriae biosynthesis [78,79]. MlrA is a DNA-binding transcriptional activator that is required for the expression of *csgD* which is responsible for the expression of the curli *csgBAC* genes in *E. coli.* RpoS activity, on the other hand, is required for the expression of *mlrA* [78,79,80]. Excision of the prophage from the *mlrA* gene was shown to restore curli biosynthesis and biofilm formation [75]. In some non-O157:H7 strains that were limited in biofilm formation and curli expression, they found that prophage insertions in *mlrA* could not be complemented. These strains shared a common phenotype that had a defect in motility. Thus, it was suggested that motility was another factor that limited biofilm formation [28]. We surmise in our studies with *E. coli* ATCC 25922 that the limited biofilm formation and Congo Red binding that we observed may similarly result from a defect in *mlrA* or RpoS, as reported for O157:H7 and other non-O157:H7 strains [28]. *E. coli* ATCC 25922 has been previously reported to be a strong biofilm producer [81,82]. It has also been found that inactivation of the *purL* gene in *E. coli* ATCC 25922 can result in limited biofilm formation [83]. This finding suggests that there may be other uncharacterized components or genetic elements that could affect biofilm formation in *E. coli* besides defects in mlrA, RpoS, or a lack of motility.

The differential growth between the *E. coli* O104:H4 and O157:H7 strains in the HBD cell culture medium was unexpected. Normally, this cell culture medium is used for detecting the activation of TLR(s) that are present in specialized HRK293 cells. HEK293-TLR cells have been used previously to evaluate the potential host innate immune response to several other live Gram-negative pathogens. The growth of these microorganisms in an HBD cell culture medium, however, was not observed [45]. In the present study, we examined the activation of TLR2 and 4, which are two TLRs that are exogenously expressed on the surface of mammalian cells [46], by live *E. coli* strains. We found that the live *E. coli* strains activated both TLR2 and 4, with TLR4 activation being more prevalent than TLR2 activation. We have previously observed that under certain conditions, TLR2 or 4 activation can vary widely in some Gram-negative pathogens. TLR2 and 4 could both be activated with one being more dominant than the other, TLR2 or 4 alone could be activated, or neither TLR2 nor 4 was activated [45]. The activation of these TLRs depends on the microorganism being examined or the conditions of growth of the microorganism [45]. Because the HBD cell culture medium was a proprietary medium (Invivogen), we can only speculate that there may have been at least two categories of inhibitory compounds (or antibiotics) present in the HBD cell culture medium. At least one of these allowed the *E. coli* O104:H4 strains to grow but not the O157:H7 strain. In comparison, in a rich, complex medium, such as the MH II medium, the growth of the *E. coli* strains was essentially identical (see Figure 3C). The first type of inhibitory compound that may have been present in the HBD cell culture medium is an antibiotic(s) to maintain the sterility of the HEK293 TLR cell line, for example, a combination of penicillin/streptomycin and/or another proprietary anti-microbial agent. The second type of inhibitory compound that may have been present is an antibiotic (or proprietary reagent), for example, ampicillin, to help maintain or select the SEAP expression plasmid in the transfected HEK293 TLR cell line [84,85]. The inhibitory reagent may also have been a combination of antibiotics or inhibitory components in the HBD cell culture medium. From our studies, we know that *E. coli* O157:H7 was relatively sensitive to streptomycin and ampicillin, while the *E. coli* O104:H4 strains we used in the present study were resistant to both antibiotics (see Appendix A) [58,59,60]. Additionally, the *E. coli* O157:H7 and O104:H4 strains that were used in the present study were resistant to penicillin G (Appendix A), which eliminate it as the differential or selective agent in the HBD cell culture medium. A mixture of HBD cell culture medium and Congo Red agar was also evaluated. The differential growth of the *E. coli* strains on a solid HBD cell culture medium and Congo Red agar was consistent with what we observed with the HBD cell culture medium. The colonies of *E. coli* O104:H4 on this solid medium, however, were black in color.

Further studies are needed to analyze the culture conditions that allow the *E. coli* O104:H4 strains to grow in the HBD cell culture medium but not the *E. coli* O157:H7 strain. Additionally, other strains of *E. coli* O104:H4 and O157:H7 as well as other non-O157:H7 strains associated with HUS must be examined to evaluate the specificity of the medium. That the *E. coli* O104:H4 strains were able to grow in flasks with an HBD cell culture medium when shaken without CO_2_ was noted. However, smaller (0.2–2.0 mL), stationary cultures (96-well plates or small tubes) of O104:H4 strains always grew with CO_2_. In addition, the effect of CO_2_ on virulence and the possible augmented expression of AAF/I fimbriae and other pathogen virulence proteins, such as Stx2a, must be investigated. Our report of new findings on the difference between *E. coli* O104:H4 and O157:H7 will further help differentiate between these two pathogens. In the future, it may assist in establishing the type of appropriate treatment for patients presenting with HUS. The involvement of alternate metabolic pathways that allow for the growth of *E. coli* O104:H4 and enhance virulence may offer potential targets for possible therapeutic intervention for HUS caused by this pathogen.

## Figures and Tables

**Figure 1 microorganisms-11-01744-f001:**
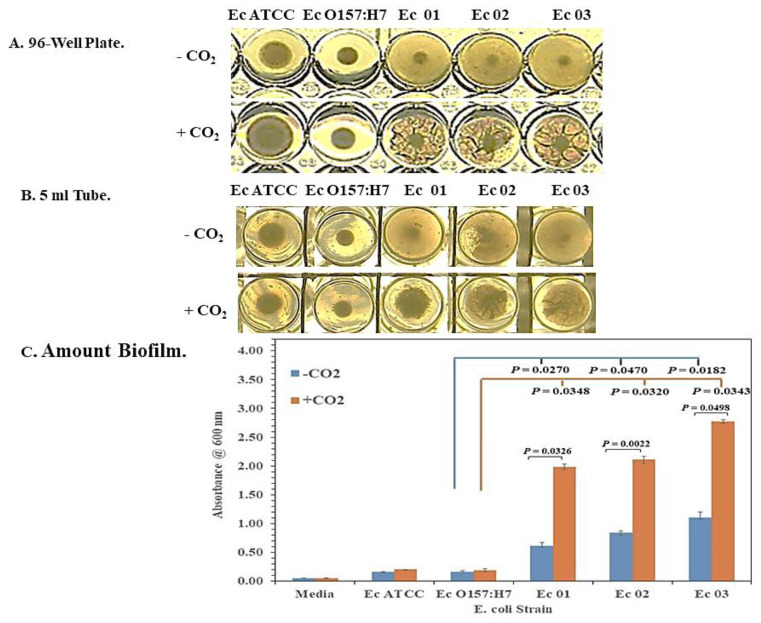
CO_2_ enhances biofilm formation by the *E. coli* O104:H4 strains but not the *E. coli* O157:H7 strains. The *E. coli* strains that were examined were *E. coli* O104:H4 (strains Ec01, Ec02, and Ec03), ATCC 25922, and O157:H7. The cultures were incubated in MH II broth overnight at 37 °C. All strains were examined in triplicate, and figures represent 1 of 3 separate studies. Only a single well of the triplicate is shown (**A**,**B**). A. Biofilm formation by the *E. coli* strains in a 96-well plate with and without CO_2_. B. Biofilm formation by the *E. coli* strains in 5 mL culture tubes with and without CO_2_. (**C**) Quantitative crystal violet analysis of the amount of biofilm formed by the *E. coli* strains in 5 mL culture tubes with and without CO_2_. In (**C**), results are presented as geometric mean with standard error of the mean. *p*-values are shown for the comparison of the amount of biofilm formation by *E. coli* O157:H7 with (orange line) and without (blue line) CO_2_ and the amount of biofilm formation by the O104:H4 strains under the same conditions. *p*-values, when significant, are also shown for comparison of biofilm formation between individual strains with and without CO_2_.

**Figure 2 microorganisms-11-01744-f002:**
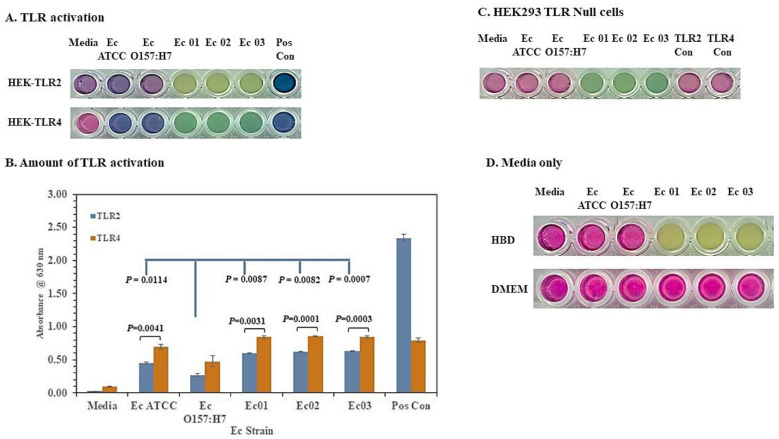
*E. coli* O104:H4 strains appear to grow in a novel cell culture medium but the *E. coli* O157:H7 strain does not. Live *E. coli* strains were examined for their ability to activate TLR2 or 4. The figure presents representative results from one of three studies we conducted with live bacterial strains after overnight incubation at 37 °C. (**A**) Bacterial strains and TLR control antigens were added in triplicate wells of HEK294 cells expressing TLR2 or 4 in an HBD cell culture medium overnight at 37 °C with 5% CO_2_ in 96-well plates. Note the color reaction difference between the *E. coli* O104:H4 strains and the ATCC 25922 and O157:H7 strains. (**B**) The figure shows the amount of TLR2 or 4 activation by live *E. coli* strains. The results are presented as geometric means with standard errors of the mean. *p*-values are also shown comparing the amount of TLR2 activated by *E. coli* O157:H7 to the amount of TLR2 activated by the other *E. coli* strains (in blue). The differences between the results of TLR4 activation by *E. coli* O157:H7 and the other strains were not significant. Statistical comparisons are also shown between the amount of TLR2 and 4 activation of the individual *E. coli* strains. (**C**) Reaction using live *E. coli* strains with HEK293 cells not expressing TLR2 or 4 (null cells). (**D**) Incubation of live *E. coli* strains in HBD cell culture medium or DMEM cell culture maintenance medium without HEK293 cells.

**Figure 3 microorganisms-11-01744-f003:**
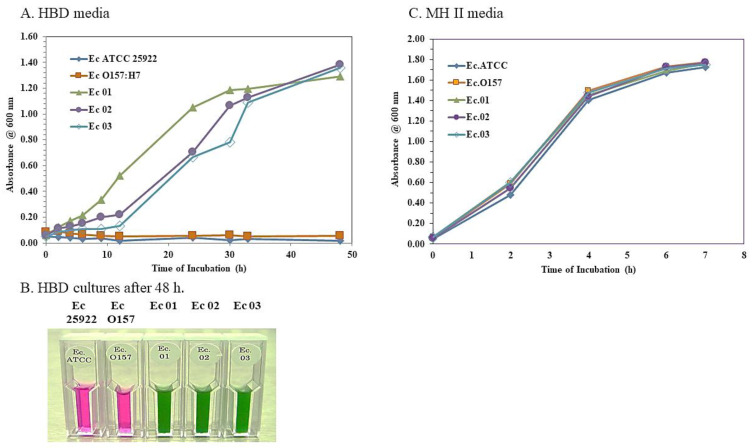
Differential growth of *E. coli* O104:H4 and O157:H7 strains in HBD cell culture medium. Single colonies of the *E. coli* O104:H4 or O157:H7 (and ATCC 25922) strains from SBA plates were used to inoculate 2 mL of MH II broth and cultures were incubated overnight at 37 °C. Overnight *E. coli* cultures were used to inoculate 12.5 mL of HBD cell culture medium (0.25 mL inoculum) or 12 mL MH II broth (0.12 mL inoculum) in 125 mL flasks and cultures were incubated at 37 °C with shaking at 180 rpm. Growth was followed by an increase in absorbance at 600 nm. The figure presents representative results from one of three independent studies. (**A**) Growth of the *E*. *coli* strains in HBD cell culture medium followed for 48 h. (**B**) Appearance of the *E. coli* cultures in HBD cell culture medium after 48 h. (**C**) Growth of the *E. coli* strains in MH II broth after 7 h of incubation.

**Figure 4 microorganisms-11-01744-f004:**
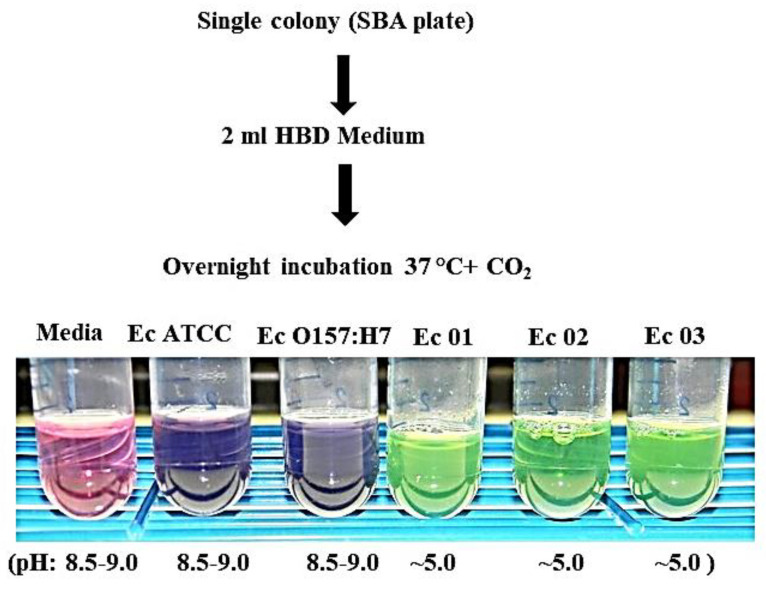
Growth of *E. coli* O104:H4 strains in HBD cell culture medium starting with single colonies. A single colony from an overnight growth on SBA plates was used to inoculate 2 mL of HBD medium and cultures were incubated overnight at 37 °C with 5% CO_2_. The figure presents representative results from one of three independent studies. There were color differences between the *E. coli* O104:H4 strains and the ATCC 25922 and O157:H7 strains. This, along with the pH differences (pH ~5.0 vs. 8.5–9.0), reflected the growth of the *E. coli* O104:H4 strains. The purple color of the *E. coli* ATCC and O157:H7 cultures may have been derived from endogenous alkaline phosphatase that came from the original inoculum.

**Table 1 microorganisms-11-01744-t001:** *E. coli* strains used in the present study.

Strains ^a^	Source/Clinical History	Phenotype/Genotype ^b^	Reference
*E. coli* ATCC 25922	Clinical isolate, Seattle, WA, 1946	sorbitol (+), β-glucuronidase (+), tellurite (−), penicillin (r), ampicillin (s), streptomycin (s), motile (+), Congo Red (±), Stx2a (−)	[58], this study
*E. coli* O157:H7	Clinical isolate, 2000	sorbitol (−), β-glucuronisase (−), tellurite (+), penicillin (r), ampicillin (s), streptomycin (s), motile (+), Congo Red (±), Stx2a (+)	this study
*E. coli* O104:H4, 2011C 3493 (Ec 01)	HUS in US 2011, traveled from Germany, May 2011	sorbitol (+), β-glucuronisase (+), tellurite (+), penicillin (r), ampicillin (r), streptomycin (r), motile (+), Congo Red (+), Stx2a (+); *stx1* (−), *stx2* (+), *eae* (−), *aatA* (+), *agg* A (+), *aggR* (+)	[59,60], this study
*E. coli* O104:H4, 2009EL 2050 (Ec 02)	Bloody diarrhea, Republic of Georgia, 2009	sorbitol (+), β-glucuronisase (+), tellurite (+), penicillin (r), ampicillin (r), streptomycin (r), motile (+), Congo Red (+), Stx2a (+); *stx1* (−), *stx* 2 (+), *eae* (−), *aatA* (+), ag*gA* (+), *aggR* (+)	[59,60], this study
*E. coli* O104:H4, 2009EL 2071 (Ec 03)	Bloody diarrhea, Republic of Georgia, 2009	sorbitol (+), β-glucuronisase (+), tellurite (+), penicillin (r), ampicillin (r), streptomycin (r), motile (+), Congo Red (+), Stx2a (+); *stx1* (−), *stx2* (+), *eae* (−), *aatA* (+), *aggA* (+), *aggR* (+)	[59,60], this study

^a^ All strains were obtained from the Critical Reagent Program at USAMRIID. ^b^ Symbols: +, positive; ±, weak; −, negative; r, resistant; s, sensitive; Stx2a, Shiga toxin 2a; *stx1*, Shiga toxin 1; *eae*, intimin; *aatA*, virulence plasmid; *aggA*, pilin subunit of aggregative adherence fimbriae I (AAF/I); *aggR*, transcription regulator.

## Data Availability

The original contributions presented in the study are included in the article/Appendix A. Further inquiries can be directed to the corresponding authors.

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
