# Peer review of "Shiga-Toxin-Producing Strains of Escherichia coli O104:H4 and a Strain of O157:H7, Which Can Cause Human Hemolytic Uremic Syndrome, Differ in Biofilm Formation in the Presence of CO2 and in Their Ability to Grow in a Novel Cell Culture Medium"

_microorganisms, 2023, doi:10.3390/microorganisms11071744_

Round 1

Reviewer 1 Report (Previous Reviewer 1)

The manuscript titled “Comparison of Shiga toxin-producing strains of Escherichia coli O104H4 and O157:H7 that can cause human hemolytic uremic syndrome uncovered differences in the response of biofilm formation to CO2 and the ability to grow in a novel cell culture medium” by Kei Amemiya et al. analyzed the differences of three  Escherichia coli serotypes O104:H4, one O157:H7 and an ATCC strain. The manuscript has notably improved in their clarity, how results are shown and discussed. However, the text still needs editing in the English language. 

In addition, since you only study one EHEC strain throughout the manuscript, and different isolates/variants could respond differently to biofilm formation or growth in the mentioned new media, you should state on the title and abstract that these results are based on only one E.coli O157:H7 isolate.

The text needs editing in the English language.

Author Response

Reviewer No. 1.

The manuscript titled "Comparison of Shiga toxin-producing strains of Escherichia coli 0104H4 and O157:H? that can cause human hemolytic uremic syndrome uncovered differences in the response of biofilm formation to CO2 and the ability to grow in a novel cell culture medium" by Kei Amemiya et al. analyzed the differences of three Escherichia coli serotypes O104:H4, one O157:H? and an ATCC strain. The manuscript has notably improved in their clarity, how results are shown and discussed. However, the text still needs editing in the English language.

In addition, since you only study one EHEC strain throughout the manuscript, and different isolates/variants could respond differently to biofilm formation or growth in the mentioned new media, you should state on the title and abstract that these results are based on only one E.coli O157:H? isolate.

  1. We have modified the title and abstract:

Title:

Shiga toxin producing strains of Escherichia coli O104:H4 and a strain of O157:H7, which can cause human hemolytic uremic syndrome, differ in biofilm formation in the presence of CO2 and in the ability to grow in a novel cell culture medium

Abstract:

                        One pathogen that commonly causes gastrointestinal illnesses from the consumption of contaminated food is Escherichia coli O157:H7. In 2011 in Germany, however, there was a prominent outbreak of bloody diarrhea with a high incidence of hemolytic uremic syndrome (HUS) caused by an atypical, more virulent E. coli O104:H4 strain. To facilitate the identification of this lesser known atypical E. coli O104:H4 strain, we wanted to identify phenotypic differences between it and a strain of O157:H7 in different media and culture conditions. We found E. coli O104:H4 strains produced considerably more biofilm than the strain of O157:H7 at 37 °C (P = 0.0470-0.0182). Biofilm production was significantly enhanced by the presence of 5% CO2 (P = 0.0348-0.0320). In our studies of the innate immune response to the E. coli strains, we used HEK293 cells that express Toll-like receptors (TLRs) 2 or 4. We found that E. coli O104:H4 strains had the ability to grow in a novel HEK293 cell culture medium while the E. coli O157:H7 strain could not. Thus, we uncovered previously unknown phenotypic properties of E. coli O104:H4 to further differentiate this pathogen from E. coli O157:H7.   

The text needs editing in the English language

-We sent the manuscript to an editor for review

Reviewer 2 Report (Previous Reviewer 2)

All the comments from the first review had been considered and answered. Major modifications have been performed. Discussion is quite long and the text, especially in discussion, was still somewhat difficult to follow. Editing process hopefully helps here. Perhaps shorter sentences and careful review of language bring the text more reader friendly.  

Title, The title could still be more focused. Proposition: Shiga toxin-producing strains of Escherichia coli O104:H4 and O157:H7 differ in biofilm formation and in the ability to grow in a novel cell culture medium

In figure 1c and especially in figure 2b, it is not clear which p-values refer to which comparisons when three or four p-values are marked within the same horizontal lane.  Please, mark p-values as in previous version of the manuscript (ie. one horizontal line for each comparison).

Abstract, line 22: It is stated that E. coli O104:H4 and O157:H7 are closely related. Is this reported before? If so, refer this in introduction or remove “closely related”.

Discussion, lines 515-517, It is stated:  ”In the initial study there was a notable difference in the substantial  amount of biofilm produced by the E. coli O104:H4 strains when compared to the EHEC 516 pathogen E. coli O157:H7 in the absence of CO2”. Instead of “notable difference in the substantial amount”, it would be better to use more descriptive terms: “x-fold more biofilm production than….”    

Discussion, lines 555-568: This text would suit better to Results.

Discussion, lines 585-601: lpf operon was discussed, but was not analysed in this study. This chapter could be shortened or even deleted.

Language still requires some editing. Some sentences are very long and distract the reader. 

Author Response

Response to reviewer No. 2

Title, The title could still be more focused. Proposition: Shiga toxin­ producing strains of Escherichia coli O104:H4  and O157:H? differ in biofilm formation and in the ability to grow in a novel cell culture medium

-We have modified the title of the manuscript as suggested by the reviewers.   In addition, we wanted to make it more interesting to non-E. coli O157:H7 investigators and to differentiate it (“in the presence of CO2”) from a previous published study.

“Shiga toxin-producing strains of Escherichia coli O104:H4 and a strain of O157:H7, which can cause human hemolytic uremic syndrome, differ in biofilm formation in the presence of CO2 and in the ability to grow in a novel cell culture medium”

In figure 1c and especially in figure 2b, it is not clear which p­ values refer to which comparisons when three or four p-values are marked within the same horizontal lane. Please, mark p-values as in previous version of the manuscript (ie. one horizontal line for each comparison).

-We modified the way the p-values were displayed and hope they are clearer as to which data was being compared with the Ec O157:H7 results.

Abstract, line 22: It is stated that E. coli 0104:H4 and O157:H? are closely related. Is this reported before?  If so, refer this in introduction or remove "closely related" .

-We removed “closely related” from the abstract

Discussion, lines 515-517, It is stated: "In the initial study there was a notable difference in the substantial amount of biofilm produced by the E. coli 0104:H4 strains when compared to the EHEC 516 pathogen E. coli O157:H? in the absence of CO2". Instead of "notable difference in the substantial amount", it would be better to use more descriptive terms: "x-fold more biofilm production than         "

-We modified this section of the Discussion as follows:

      In our observations on the biofilms produced by the EHEC/EAEC pathotype E. coli O104:H4 we found how biofilms produced by the O104:H4 strains can be affected by the presence of CO2.  In the initial study there was a notable 3 - 7-fold difference in the substantial amount of biofilm produced by the E. coli O104:H4 strains when compared to the EHEC pathogen E. coli O157:H7 in the absence of CO2.   We then saw a significant enhancement (10 - 15-fold) of the amount of biofilm formed by E. coli O104:H4 when grown in the presence of CO2, while we observed no response by the O157:H7 strain. 

Discussion, lines 555-568: This text would suit better to Results.

-Yes, it could have been placed in the Results section with Fig. S2 possibly presented as Fig. 5.  But since we made the sequence of the AAF/I genes (aggABCD) as a Supplement Fig. S2, we thought that it would be more suitable to discuss the sequences and genes in the Discussion section.  We also did not directly study the involvement of the AAF/I fimbriae or its genes in the Ec O104:H4 strains.

Discussion, lines 585-601: lpf operon was discussed, but was not analysed in this study. This chapter could be shortened or even deleted.

-We shortened the section on the lpf operon.

Comments on    Language still requires some editing. Some sentences are very the   Quality of       long and distract the reader.

English

Language

-We have sent the manuscript with our changes for editing and input.

This manuscript is a resubmission of an earlier submission. The following is a list of the peer review reports and author responses from that submission.

Round 1

Reviewer 1 Report

The manuscript titled “A novel culture medium differentiates growth between the two Shiga toxin-producing Escherichia coli serotypes O104:H4 and O157:H7 that cause human hemolytic uremic syndrome” by Kei Amemiya et al. explored the differences on antibiotic resistance, biofilm formation, TLR2 and 4 activation and growth in several media of three  Escherichia coli serotypes O104:H4, one O157:H7 and an ATCC strain. The results presented here are interesting and might advance in the differentiation of these serotypes. However, several questions still need to be addressed.

1- The title states that the present work explores two Shiga toxin-producing Escherichia coli serotypes O104:H4, however the results shown here are about three  O104:H4 strains.

2- lines 13-26: Please, state clearly the objective of the present study

3- lines 30-86: Authors should provide an introduction on TLR2 and 4, and why it is included in the present study.

4- lines 30-34: “Pandemics caused by microorganisms may occur every generation, such as the influenza pandemic of 1918 or coronavirus pandemic of 2019. Between pandemics, however, there are numerous smaller outbreaks that occur annually One pathogen most frequently associated with such outbreaks is Escherchia coli O157:H7, which is a Shiga toxin-producing E. coli (STEC) that is associated with consumption of contaminated food [1,2].”

Authors suggest that a STEC pandemic could be possible. However, due to the nature of the infection it would be rare that it happens. Please, provide a reference or rephrase the statement.

5- lines 41-44: “One such element was lysogenic phages that carrying genes for Shiga toxin (Stx), which is also called vero toxin, that closely resembles the toxin expressed by Shigella dysenteriae 1, and another genetic element was plasmids that carried virulence factors [12-15].” 

Sentence a bit confusing, please rephrase.

6- lines 48-49: “Shiga toxins, like ricin, are AB toxins, with ricin composed of one A subunit and one B 48 subunit, while Shiga toxins are composed of one A subunit linked to five B subunits 49 [16,20,21].” 

It is not clear in this sentence why you mentioned ricin.

7- line 91: Could you provide more data on the E. coli O157:H7 used in the present study?

8- lines 144-6: “To the dry pellet, 1.0 ml of 95% ethanol was 144 added and sample vortexed vigorously to dissolve the pellet and samples left for 15-30 145 min. In some cases, samples were left in the solvent overnight to ensure complete “

Authors said that in some cases samples were left overnight in 95% ethanol in order to dissolve the pellet completely. How did you make sure that the solvent didn’t evaporate, thus concentrating the sample?

9- lines 151-5: “ The TLR 151 expressing HEK cells were also transfected with embryonic alkaline phosphatase reporter 152 protein that is secreted (SEAP) into the medium in which its’ expression was under the 153 control of multiple NF-κB and AP-1 binding sites in the enhancer region of a SEAP gene 154 (Invivogen).”

Please, rephrase this paragraph.

10- lines 242-4: “Also, of the antibiotics that we examined, there was no instance where E. coli O157:H7 was resistant to an antibiotic”

Could you please check this sentence? On the paragraph above this (lines 237-8) you said: “E. coli O104:H4 strains and O157:H7 shared resistance (MIC 4.0 –32 µg/ml) to sulfamethoxazole, penicillin G, and partially to novobiocin (Table 2).”

11- lines 251-254 and figure 1A, top-panel: “E. coli O157:H7 251 and the control E. coli ATCC 25922 had well defined circular pellets, while E. coli O104:H4 252 strains Ec 01, Ec 02, and Ec 03 had homogeneous pellets that covered the bottom of the 253 wells.” 

You said that Ec 01, 02 and 03 had homogenous pellets, however from Fig 1 A the pellet form Ec 02 looks different form Ec 01 an 03.

12- lines 319-321: “ Significant values are also shown compared to media without and with CO2, respectively: *P ≤ 0.05; **P ≤ 0.01; ***P ≤ 0.001; ****P ≤ 0.0001; *****P≤0.000001. Statistical comparisons are also shown with individual strains between TLR2 and TLR4 activation”

Please, clarify since it is not clear.

13- lines 362-7: “reached an A600 of 1.29–1.38, and we detected 1.6x108 – 4.3x108 colony forming units (CFUs) from the E. coli O104:H4 cultures. The E. coli O104:H4 Ec 03 strain appeared to grow slightly slower than the E. coli O104:H4 Ec 01 and Ec 02 strains at the beginning, but  it reached close to the same density as the other two E. coli O104:H4 strains after 48 h. We observed no change in culture density or recovered CFUs from the E. coli ATCC 25922 or O157:H7 cultures after the same time period.”

It is not clear from this paragraph and Methods section how you determined CFUs, please clarify.

14- lines 398-400: “The color change in the 398 E. coli ATCC 25922 and O157:H7 cultures may be from alkaline phosphatase activity from 399 bacterial cells in the inoculum [55,56].”

15- lines 455-69: Authors provide a body of evidence of the response to changes in CO2 levels in the expression of virulence factors required to successfully infect a mammalian host: However, the link with the present study is weak, since they do not show anything about infection. Could you please clear up your idea?

16- Results on TLR2 and 4 activation are not discussed.

17- The present work could be improved if authors go deeper into the mechanisms that justify the differences observed in growth of E. coli O104:H4.

Reviewer 2 Report

General comments

The manuscript describes selected phenotypic characteristics of four stx-positive E. coli isolates: one O157 and three O104. The focus of the manuscript remained vague. According to the title, the focus should have been in a novel growth medium, but the manuscript also consisted of describing biochemical characteristics, antimicrobial sensitivity testing, measuring of biofilm formation, and Toll like receptor activation studies.

Apparently, the idea was to compare O104 strains related to outbreaks in Germany and in Republic of Georgia to one O157 strain. The isolate Ec01 was from a patient who travelled to Germany during the 2011 outbreak, but it was not described whether the strain was the outbreak strain or not. Also, Ec 02 and Ec 03 were isolated during the outbreak, but not verified as being the outbreak strain. No information on virulence profiles were given. For O157 strain, no background information was given at all. Results based on only four isolates with limited background information are somewhat arbitrary.

Some of phenotypic tests, such as use of different carbon compounds did not bring much additional value or at least, the results were not discussed. Eg. sorbitol fermentation difference between O157 and non-O157 are well known. To my understanding, new information was provided on two issues: 1) ability of O104 to grow on HBD medium, 2) the effect of CO2 to more abundant biofilm formation in O104. It remained unclear if more pronounced TLR4 activation in O104 than in O157 was new information or not, and what was the importance of TLR4/2 activation in these STEC strains. The text was sometimes difficult to understand, and new information was difficult to extract from information what is already known. A careful review of language and modification of introduction and discussion, and perhaps a shorter length of the manuscript would help to focus the text. Figures were very informative.

The manuscript contains 75 references. Out of all 75, only two original articles and one website reference dated from last five years. All other references were older. An updated literature search on the topics covered is warranted.  

Spesific comments

Lines 12-14: The first sentence in abstract is not correct. For instance, Campylobacter and Salmonella are far more common in food borne infections than E. coli O157.

Lines 22-23, lines 308-310: The higher activation of Toll like receptor 4 than TLR2 was mentioned in abstract, and therefore considered one of the main results. However, the results were only mentioned in one sentence in chapter 3.3 (line 308-310, and Fig 2B), which was about growth of strains in HDB-medium, not about TLR activation. The results of TLR activation were not discussed, nor was any background information on TLR given in introduction. In Fig 2B, it is mentioned that statistical analysis (marked with astericses) on TLR activation was compared between E. coli strains and media. Why comparison was not between Ec O157 or Ec ATCC and O104 strains? Is it true that there was no significant difference in TLR4-production between media and of Ec O157 (ie. no asterics aboveTLR4 column)? Division of significance values in five different categories is excessive. Two categories would be enough. Also, the sentence in abstract (lines 22-23) was unclear when compared to figure 2B. “TLR2/4 activation was similar with all live E. coli strains examined with significantly more TLR4 activated than TLR2 except for E. coli O157”.

Lines 172-175: How was the background color adjusted in plate reading? Was the absorbance, depicted in Fig 2B, calculated by subtracting the absorbance of HEK293 TLR Null cells from HEK293 TLR transfected (Fig 2A-Fig 2C)? Please, add this information to the text.

Lines 293 and 354: Chapters 3.3 and 3.4 have almost same titles. These chapters, concerning growth of bacteria in HBD medium, can be combined. TLR activation could be explained in a separate chapter.

Fig1A and 1B, Supplementary Fig, and chapter 3.2 (lines 248-259, lines 290-291): Is it possible that the different appearance of pellets of O157 and O104 strains in 96 well plates and tubes are partly because of autoaggregation and not entirely because of biofilm formation? Shciller et al (Virulence 2021, VOL. 12, NO. 1, 346–359) has studied the effect of different aggregative adherence fimbriae (AAF) types on aggregative adherence and biofilm formation on E. coli O104 background. That study concluded, that autoaggregative strains produce less biofilm, but mediate stronger adherence to host cells. This publication should be referred and discussed here as well. Is it possible to determine which AFF types the strains used in your study represent?

Lines 412-432: The ability of O104 strains to grow on HDB is interesting. Is it only an observation or would you like to discuss whether this could be tested as selective medium for non-O157 E. coli?

Lines 117 and Table 1: Was the monoclonal anti-Shigatoxin antibody directed towards the subunit A of the toxin (as suggested in text in line 117) or was it against the stx-subtype 2a (as suggested in Table 1)? Anyway, it would be good to determine the stx subtype of the strains as it is known that stx2a subtype is associated to HUS.